# Development of New Red-Fleshed Seedless Table Grapes: In Vitro Insights on Glucose Absorption and Insulin Resistance Biomarkers

**DOI:** 10.3390/foods14234035

**Published:** 2025-11-25

**Authors:** Ana Belén Bautista-Ortín, Alejandro Martínez-Moreno, Ana Leticia Pérez-Mendoza, María José Carrasco-Palazón, Lucía Osete-Alcaraz, Laura Soriano-Romaní, Elena Díez-Sánchez, Juan Antonio Nieto, Sonia Soto-Jover, Encarna Gómez-Plaza

**Affiliations:** 1Department of Food Science and Technology, Faculty of Veterinary Sciences, University of Murcia, 30100 Murcia, Spain; anabel@um.es (A.B.B.-O.); martinezmoreno@gmail.com (A.M.-M.); analeticia.perez@um.es (A.L.P.-M.); mariajose.carrasco1@um.es (M.J.C.-P.); lucia.osetea@um.es (L.O.-A.); 2AINIA, 46980 Valencia, Spain; lsoriano@ainia.es (L.S.-R.); ediez@ainia.es (E.D.-S.); janieto@ainia.es (J.A.N.); 3Bloom Fresh International Ltd., 70-78 York Way, Unit A, London N1 9AG, UK; sonia.soto@bloomfreshglobal.com

**Keywords:** table grapes, red-fleshed, seedless, phenolic compounds, caco-2, glucose, uptake, insulin resistance

## Abstract

There is increasing interest in foods that support both physical and psychological health. Red and black fruits are notable for their high phenolic content and associated biological activities. However, their natural sugar content may raise concerns regarding glycemic impact. Recent breeding programs have developed new seedless table grape varieties with black skin and red pulp, aiming to enhance phenolic content and reduce glycemic response. This study evaluates these novel grape varieties using in vitro models of intestinal absorption and hepatic insulin resistance. Specifically, we assessed phenolic content, antioxidant capacity, glucose transport across intestinal cells, and the modulation of biomarkers related to insulin resistance. The results showed that the new grape varieties (hybrids) showed total phenolic contents of 52.4–187.3 mg GAE/100 g FW and antioxidant capacities ranging between 195.3 and 762.7 mg Trolox equivalents/100 g FW, both higher than those of commercial table grapes. These new varieties also showed a lower percentage of intestinal glucose transport than commercial grapes and pineapple in caco-2 cells, suggesting an improved regulation of glucose uptake. Theoretical transport values confirmed a reduced glycemic impact for most hybrids, while absorbed fractions of RF03, RF05, and RF06 also restored hepatic glycogen levels under insulin-resistant conditions, indicating enhanced glucose metabolism. Overall, our in vitro findings suggest that these new grape varieties may help modulate postprandial glucose levels, supporting their potential as a healthier fruit option.

## 1. Introduction

Nowadays, there is a growing concern among consumers about their health and well-being, which has led to an increasing demand for foods that can improve both physical and psychological health. Among these foods, fruits have attracted great interest due to their benefits for physiological well-being and their potential to prevent pathological disorders.

Fruits rich in polyphenolic compounds—such as anthocyanins, tannins, flavonols, stilbenes, and phenolic acids—provide an additional benefit through their strong antioxidant activity. For this reason, red fruits and pomegranates are often classified as “superfruits.” For example, blueberries (*Vaccinium* spp.) have gained significant attention because of their high content of bioactive molecules, especially anthocyanins, the compounds responsible for their characteristic purple-blue color [1,2,3].

In the case of table grapes, seedless varieties are the most popular due to their ease of consumption. Their polyphenolic content and profile vary significantly among cultivars [4,5], and are influenced by several factors such as geographical origin, agroclimatic conditions, and soil characteristics [6,7]. In general, table grapes—particularly seedless ones—contain lower levels of phenolic compounds compared to wine grapes and other red fruits [8]. White grapes have lower phenolic content than red grapes, mainly because anthocyanins are present in the skins of the latter [8,9].

In white grapes, the main phenolic compounds are monomeric catechin, epicatechin, polymeric flavanols, and various phenolic acids, especially gallic acid [9,10]. In contrast, red grapes are characterized by higher concentrations of anthocyanins—responsible for the red color of their skins—and flavanols [11]. The pulp of both white and red grapes contains only small amounts of phenolic compounds, primarily catechins, epicatechins, and phenolic acids [7,12,13]. Other phenolic compounds, such as flavonols and stilbenes, are also present in small quantities, mostly in the skins [14,15].

Phenolic compounds exert a wide range of biological effects in vitro. Many of these effects are related to the ability of polyphenols to protect against oxidative stress caused by free radicals. Since oxidative stress plays a key role in the development of chronic diseases [16], consuming fruits rich in phenolic compounds could help reduce the risk of such conditions [17,18,19,20,21,22,23].

However, fruit consumption is sometimes limited by their high sugar content, which increases blood glucose levels and stimulates insulin secretion by the pancreas. Insulin promotes the translocation of glucose transporters, allowing glucose to enter cells and serve as an energy source. In liver cells, insulin stimulates glycogenesis, favoring glycogen production from absorbed glucose. During fasting, this glycogen is degraded, releasing glucose into the bloodstream to maintain normal blood levels [20,21]. In cases of insulin resistance, however, tissues respond poorly to insulin, leading to impaired glucose regulation [21]. Because grapes typically contain more sugar than other fruits, some consumers perceive them as less healthy.

Recent studies suggest that phenolic compounds may limit glucose bioavailability through two main mechanisms. First, they can inhibit carbohydrate digestion by interacting with proteins, polysaccharides, and metals, thereby affecting the activity of digestive enzymes such as α-amylase and α-glucosidase through non-covalent interactions [22,23]. Second, phenolic compounds may reduce intestinal glucose absorption. Glucose uptake is higher when glucose is consumed in water than in fruit juice [24]. Phenolic extracts from *Hibiscus sabdariffa*, rich in anthocyanins, can attenuate postprandial glycemia and improve diabetes-related metabolic markers [25]. These effects may involve modulation of intestinal glucose transporters SGLUT1 and GLUT2 [26].

Since evidence indicates that fruits with high phenolic content can combine antioxidant effects with improved glucose regulation and a lower glycemic index, breeding programs have been developed to create new table grape varieties rich in phenolic compounds. These varieties typically have red skins and red pulps, similar to teinturier grapes.

This study focuses on several of these new varieties, which are not yet available on the market. Their phenolic content and antioxidant activity were evaluated. Furthermore, the most promising varieties were analyzed in vitro to assess their potential to reduce glucose uptake during intestinal absorption and their effects on risk factors associated with diabetes, such as biomarkers related to insulin resistance.

## 2. Materials and Methods

### 2.1. Grape and Sample Preparation

The new table grape varieties were obtained using the classic hybridization technique from different genetic materials existing in Bloom Fresh International Limited. Numerous crosses are made to obtain seedlings with the desired properties: seedless and teinturier pulp. Hybridization involves the following stages: (i) the emasculation of the flowers of the female parent one week before flowering and bagging of bunches to avoid pollination with ambient pollen, (ii) the collection, conditioning and conservation of pollen from the male parents in flowering, and (iii) pollination several times while viable stigmas remain, removing and replacing the bag at each pollination and keeping the bunch protected with the bag until it is fully fertilized. Two existing varieties were selected to create a new one with different characteristics.

Wine grapes have been used for the introduction of the genetics responsible for the teinturier pulp. In cases where the female parent is seedless, the embryo rescue technique was used. Once the embryo was sufficiently mature, it was planted in test tubes, then transplanted to a greenhouse and finally established in the field to obtain a new accession. Of those obtained, only 12 (RF01-RF12) were selected due to their good agronomic performance, in terms of productivity, agronomic management and good post-harvest quality after 45 days of refrigerated storage at 1–2 °C, and for their sensory attributes, such as berry size (diameter larger than 18 mm), firm and crisp texture, and good flavour (balanced sugar/acidity), and their concentration and phenolic composition was compared with the commercial grape varieties: Timpson (white variety), Krissy (red variety) and Melody (black variety). All grape samples were harvested at their optimum maturity level to be consumed (19 °Brix) and stored at −18 °C until analyses.

The sample preparation was carried out using 60 g of defrosted berries from each hybrid and commercial variety and then crushed in a grinder. A total of 10 g of the homogeneously crushed berries were combined with 40 mL of an ethanol/water solution (80%, *v*/*v*) in falcon tubes and in triplicate. The samples were shaken at 250 rpm in an orbital shaker at room temperature and in darkness for 20 h. Then, the samples were centrifuged at 8000 rpm for 20 min and the supernatant volume for each sample was adjusted to 50 mL using a volumetric flask. After that, the extracts were filtered through 0.45 µm nylon membrane filters and maintained at −18 °C until analysis to slow down the degradation and oxidation of phenolic compounds. Moreover, the extracts were tempered before carrying out the analyses to promote greater solubility of the phenolic compounds.

The roadmap of the sample preparation is shown in Figure 1.

### 2.2. Total Phenolic Content (TPC)

The concentration of total TPC in the extracts obtained from different grape samples was determined according to the Folin–Ciocalteu method [27] with some modifications. Briefly 500 µL of the sample extract was mixed with 4000 µL of water and 500 µL of Folin–Ciocalteu reactive. Then, the sample is mixed in the vortex and 500 µL of 10% (*w*/*v*) sodium carbonate was added after 5 min. The sample was agitated and incubated for 1 h at room temperature in darkness. At the end of the incubation period, absorbance was measured using a UV–Vis spectrophotometer (Evolution 300, Thermo Fisher, Madrid, Spain) at a wavelength of 765 nm. A standard calibration curve was prepared with gallic acid and the results were expressed as mg gallic acid equivalents (mg GAE) by 100 g of fresh weight (FW).TPC (mg/100 g) = [(Absorbance _765 nm_ − 0.067)/0.006] × 0.05 × 10

### 2.3. Total Anthocyanins

The concentration of anthocyanins (TA) is obtained by measuring the absorbance at 520 nm of the extract diluted 1:25 (*v*/*v*) in 0.1 N HCl after 30 min according to [28].TA (mg/100 g) = Absorbance _520 nm_ × 22.76 × Dilution Factor × 0.05 × 10

### 2.4. Total Tannins

Methyl cellulose precipitable tannins (TT) were determined by the methyl cellulose precipitation method [29]. Briefly, 200 μL of extract and 600 μL of a solution of methylcellulose (0.04%) were put in 2 mL Eppendorf and after 3 min, 400 μL of a saturated solution of ammonium sulfate and 800 μL water were added. In the control, the 600 μL methylcellulose was replaced by water, which was used to determine the absorbance corresponding to the tannins. The sample was stirred and allowed to stand. After 10 min, the samples were centrifuged at 10,000 rpm for 5 min and the absorbance at 280 nm was measured. The results were expressed as mg (−)-epicatechin equivalents by 100 g of FW.TPC (mg/100 g) = [(ΔAbsorbance _280 nm_ − 0.0108)/0.0131] × 0.05 × 100

### 2.5. Antioxidant Capacity

The antioxidant capacity (TEAC) was measured based on the abilities of different substances to scavenge the ABTS+ radical cation generated by filtering an ABTS solution through manganese dioxide powder compared with a standard antioxidant (Trolox, Sigma, St. Louis, MO, USA) in a dose–response curve using the method proposed by [30]. Finally, AC of the samples was expressed as mg Trolox equivalents by 100 g of FW (mg TE/100 g FW).TEAC (mg/100 g) = [(%ΔA _750 nm_ − 0.715)/294.97] × Dilution Factor × 250 × 0.05 × 100

### 2.6. Determination of Phenolic Compounds by HPLC

#### 2.6.1. Anthocyanins and Flavonols

The separation of anthocyanins and flavonols present in 3 µL of extracts obtained from different grape samples was carried out in a Waters Acquity Arc liquid chromatograph (Waters, Milford, MA, USA) equipped with a Waters 2998 diode array detector (Waters, Mildford, MA, USA). The column was maintained at 55 °C was a Poroshell120 EC-C18 core–shell column (150 mm × 2.1 mm, 2.7 µm, Agilent Technologies, Santa Clara, CA, USA). The separation method used was according to [31], using as mobile phases 1% formic acid in water (A) and 1% formic acid in 1:1 (*v*/*v*) methanol/acetonitrile (B) at a flow rate of 0.3 mL/min. The elution conditions were 100% A for 2 min, lineal increase from 0 to 15% B in 33 min, from 15 to 21% B in 15 min, and from 21 to 30% B in 20 min followed by the washing and re-equilibration of the column.

Compounds were identified by comparing their UV spectra recorded with the diode array detector with those reported in the literature and confirming each peak identity also using a QDA mass detector (Waters, Pittsburgh, PA, USA). The mass spectrometer operated in positive-ion mode for anthocyanin confirmation, with a capillary voltage of 1.5 kV and in negative-ion mode for flavonol confirmation, with a capillary voltage of 0.3 KV. In both modes, the cone voltage was 30 V, and the desolvation temperature was 350 °C. Mass scans (MSs) were measured from *m*/*z* 100 up to *m*/*z* 1200.

The anthocyanins were quantified at 520 nm as malvidin-3-glucoside chloride (Extrasynthese, Gernay, France) using a range of 5–600 mg/L (R^2^ = 0.9999). The LOD and LOQ were in the ranges of 0.06–0.20 mg/L (LOD) and 0.20–0.60 mg/L, respectively. The intra-day and inter-day precision (%RSD) were lower than 6.2% and 8.5%, respectively, and the recoveries ranged between 91.6% and 119%. The flavonols were quantified at 360 nm as quercetin-3-glucoside (Extrasynthese, Gernay, France) using a range of 0.5–50 mg/L (R^2^ = 0.9998). The LOD and LOQ ranged from 0.09 to 2.09 mg/L and 0.56–8.19 mg/L, respectively. The intra-day and inter-day precision (%RSD) was less than 3.2% and recoveries ranged between 91.2 and 98.7%.

#### 2.6.2. Stilbenes

The method described by [32] was used for the extraction and analysis of stilbenes by HPLC-DAD. Briefly, 5 mL of extract was mixed with 5 mL of ethyl acetate. After, the samples were homogenized using an Ultraturrax T-25 (Janke & Kunkel, Ika-Labortechnick, Breisgau, Germany) and stirred at 200 rpm for 20 min. Then, the samples were centrifuged at 5000 rpm at 4 C for 5 min and the organic phase was dried in a Centrivap concentrator (Labconco, Kansas City, MO, USA) and re-diluted in 2 mL MeOH for their analysis. The analysis was carried out in a Waters 2695 HPLC system (Waters, Milford, MA, USA) with a Waters 2996 photodiode array detector at a temperature of 30 °C. The separation of the 20 µL sample was performed on a Lichro Cart RP-18 column (Merck, Darmstadt, Germany), 25 × 0.4 cm, 5 μm particle size, using water and 5% of formic acid (solvent A) and acetonitrile (solvent B) at a flow rate of 1 mL/min. The elution conditions were as follows: 0% B for 0 min; 15% B for 15 min; 20% B for 40 min; 55% B for 30 min; and 0 B for 32 min. The quantification of these compounds was carried out at 306 nm using the corresponding standard purchased from Sigma (Sigma-Aldrich, Madrid, Spain) using a calibration range of 0.25–50 mg/L (R^2^ > 0.9995). The LOD and LOQ were as low as 0.003 mg/L and 0.005 mg/L, respectively. The intra-day precision was 0.5–7.2% and the inter-day precision was up to 16%, and recoveries ranged between 94.2 and 103.5%.

### 2.7. Simulation of In Vitro Gastrointestinal Digestion

The simulated in vitro gastrointestinal digestion process was conducted using DigestSim, a dynamic digestion system developed by AINIA. This equipment allows us to replicate the upper gastrointestinal tract of human adults. This system is a computer-controlled setup with two modules: one emulates gastric digestion, and the other simulates intestinal digestion.

The gastrointestinal digestion process was conducted based on the protocol of [33] with few modifications. Prior to gastric and intestinal digestion, an oral digestion step was performed. For this purpose, 150 g of fresh grapes were selected, each time taking a grape from a different bunch, trying to select different sizes to generate adequate dispersion. The sample was mixed with a simulated salivary solution at 37 °C consisting of an electrolytical solution with citrate buffer (pH 6), containing 9600 α-amylase units [34]. The pH was immediately adjusted to a basic value (6.5–7) and the mixture was stirred during 2 min at 500 rpm and 37 °C, allowing us to generate the oral digest. After that, the oral digested were placed into the gastric chamber, which contained 5 mL of simulated gastric solution, consisting of an electrolytic gastric solution containing pepsin (0.3 mg/mL) and gastric lipase solution (0.2 mg/mL) preheated at 37 °C. Also, the intestinal chamber was loaded earlier with 60 mL of an intestinal secretion to initiate the whole assays. Gastric and intestinal secretions were freshly prepared based on [35]. During gastric digestion, the pH was continuously controlled by adding a 1 M HCl solution, according to a physiological pH change during gastric digestion (from pH 5 to pH 2), whereas the small intestine (pH 6.5–7) was maintained at pH 7 using a 1 M HCO3 solution. For gastric digestion, a pepsin solution (0.3 mg/mL) was continuously added for 2 h at a rate of 0.5 mL/min, while gastric emptying was conducted for 3 h following an in vivo physiological gastric emptying batch reported by [35]. Simultaneously, 7% (*w*/*v*) of pancreatin solution (pancreatic amylase, lipase, ribonuclease, and trypsin) and 2% (*w*/*v*) bile solution were added at a flow rate of 0.5 mL/min throughout the experiment for 6 h, as well as the intestinal emptying [35].

To emulate the further intestinal absorption process, samples were collected every hour from the intestinal chamber, immediately placed in a precooled sealed bottle at −20 °C, and were maintained at this temperature until further studies. All the collected samples for each digestion were placed in the same bottle, therefore, mixing the different picked aliquots. This process allowed us to accumulate the intestinal digested material, represented by the intestinal digested at different digestion times.

The in vitro digestion simulation was carried out in duplicate for accuracy. The same process using water as a control, representing the blank digestion with the absence of the sample was also conducted.

### 2.8. Cells

Human epithelial cell line Caco-2 obtained from the American Type Culture Collection (ATCC^®^ HTB-37^TM^, Manassas, VA, USA) was used to carry out intestinal transport studies. For its maintenance, cells were cultured in growth medium containing Eagle Minimum Essential Medium (EMEM, ATCC) supplemented with 20% fetal bovine serum (FBS, Gibco-Thermo Fisher Scientific) and antibiotics (penicillin 100 U/mL, streptomycin 30 μg/mL, PAN-Biotech, Aidenbach, Germany), following ATCC recommendations. A biocompatibility test was carried out with the samples to determine the maximum biocompatible concentration with the cell model and then, these cells were used in the intestinal absorption model.

In addition, a cell line exhibiting epithelial-like morphology that was isolated from a hepatocellular carcinoma was obtained from ATCC (HepG2, HB-8065^TM^). Cells were cultured following ATCC recommendation in EMEM supplemented with 10% FBS and antibiotics. These cells were used in the insulin resistance model.

### 2.9. Cell Viability Measurement

Samples used in the different cell models were the soluble fractions (SFs) generated from the simulated gastrointestinal digestion. A blank adding water instead of the sample was generated as a control digest. These SFs were filtered and frozen until used. Two replicates of each product obtained in the bioaccessibility studies were integrated to perform cellular assays. A cytotoxicity assay was performed with serial dilutions of the samples in Caco-2 using alamarBlue Cell Viability Reagent (Thermo Fisher Scientific, Waltham, MA, USA). After the exposure period (60 min), reagent was added according to manufacturer instructions and fluorescence was read at Ex 530 nm Em. 590 using Fluoroskan^TM^ FL (Themo Fisher Scientific). Dimethyl sulfoxide (DMSO) at 20% was used as a positive cytotoxicity control. Percentage of cell viability relative to control cells (untreated) was determined as follows:Cell viability (%) = (Fluorescence Units from the sample/Fluorescence Units from the control) × 100.

### 2.10. Transepithelial Intestinal Transport Studies and Glucose Levels

The entire accumulated intestinal digest was homogenized and subjected to a 10 min heat treatment in a water bath at 90 °C to inactivate digestive proteases. Afterward, the sample underwent centrifugation to separate it into a soluble fraction and a non-soluble fraction, representing the components that did not dissolve during digestion. This non-soluble fraction contains potentially non-absorbable digest material that passes through the small intestine to reach the large intestine. The SFs were used in transport studies.

Intestinal absorption through intestinal epithelial cells was assessed after optimizing culture conditions. Briefly, Caco-2 cells were seeded into polyester membrane (0.4 µm of pore size; Corning, Somerville, MA, USA) inserts and cells were incubated at 37 °C during 21 days for its differentiation. The integrity of cell monolayers was determined by measuring transepithelial electrical resistance (TEER) using Millicell ERS-2 (Millipore Corporation, Burlington, MA, USA) before and after treatment. After incubation with SF of digested varieties and controls for 60 min at 37 °C, basal medium was collected obtaining the absorbed media. Both fractions were used for further treatments of macrophage and lymphocyte cell models. Only cell monolayers with TEER values higher than 300 Ω cm^2^ were considered.

For the determination of glucose levels in the absorbed media, high-performance liquid chromatography (Agilent Technologies) was performed using evaporative light scattering detector (ELSD) and a quantification limit of 5 μg/mL.

### 2.11. Liver Model to Study Modulation in Insulin Resistance Biomarkers

To evaluate the effect on insulin resistance biomarkers, the culture conditions of HepG2 were modified to induce insulin resistance in vitro following methodology previously used in this cell type [36,37]. To study the effect of the bioactive compounds, present in these samples on the sensitivity of hepatocytes to insulin, the proposed model is based on stimulating the cells with insulin and measuring the amount of glycogen that accumulates inside. In this model, the samples of interest are added and after 24 h the cell lysates are collected, either to analyze intracellular glycogen levels or to evaluate changes in the gene expression level of an intermediary in the metabolic pathway, Akt, an enzyme that belongs to the superfamily of serine/threonine protein kinases.

### 2.12. Intracellular Glycogen Level Determination

For intracellular glycogen level quantification, first total protein content was determined. The Bradford reagent (Bio-Rad Laboratories, Hercules, CA, USA) was used to quantify total protein, in which cells are lysed in cell lysis buffer (Sigma-Aldrich) and a cocktail of protease inhibitors (Thermo Fisher Scientific), and absorbance is measured at 540 nm. The results were extrapolated with the values obtained in a standard line with a known amount of protein, thus calculating the protein concentration after each treatment, following the manufacturer’s instructions. These values serve to normalize the values obtained for intracellular glycogen. A Glycogen Assay Kit (Sigma-Aldrich) was used following its instructions. The results are extrapolated in a standard line made with known glycogen concentrations and normalized with the total protein quantified.

### 2.13. RNA Isolation and Quantitative Real-Time Polymerase Chain Reaction (qRT-PCR)

After different cell treatments, cellular RNA was isolated and purified using MAXWELL equipment (Promega, Madison, WI, USA). RNA quantity and quality was measured by Nanodrop (Thermo Fisher Scientific) before cDNA synthesis. cDNA was obtained from 1 μg of quality RNA using High-Capacity cDNA reverse transcription kit (Applied Biosystems, Foster City, CA, USA). Real-time PCR was performed with TAQMAN^TM^ fast advanced master mix (Thermo Fisher Scientific) from cDNA using commercial primers of selected biomarker akt (Assay ID Hs00920503_m1) and the housekeeping β-actin (4326315E). All primers were obtained from Thermo Fisher Scientific. Quantification of gene expression was carried out in a relative manner, therefore the magnitude of physiological changes in every biomarker gene was obtained in comparison with the reference gene β-actin. For calculations, the formula 2^−ΔΔCT^ was used.

### 2.14. Statistical Analyses

Results were presented as mean ± standard error of the mean (SEM). The statistical package Statgraphics Centurion XVI.3 (StatPoint Technologies, Inc., The Plains, VA, USA) was used for the execution of a one-way analysis of variance (ANOVA) followed by an LSD (least significant difference) test (*p* ≤ 0.05) for spectrophotometric and HPLC data, while statistical significance between different conditions for in vitro assay data was assessed using Student’s *t*-test with a Welch’s correction applied in case of significantly different variances (F test) using GraphPad Prism software 8.3.1 (*p* ≤ 0.05).

## 3. Results and Discussion

### 3.1. Phenolic Compound Content and Antioxidant Activity

The results of the phenolic compound content and antioxidant activity of the different grape samples are shown in Table 1. Among the commercial grape varieties, Melody^TM^ (black skin, colorless pulp) exhibited the highest total phenolic content (61.0 mg/100 g FW), mainly due to its higher anthocyanin and tannin levels. Krissy^TM^ (red) presented intermediate values (29.8 mg/100 g FW), while Timpson^TM^ (white) showed the lowest (21.1 mg/100 g FW), a result associated with the absence of anthocyanins, although tannin levels were similar to those of Krissy^TM^ (38.6 and 33.6 mg/100 g FW, respectively). Despite these differences in phenolic content, no significant variations in antioxidant activity were observed among the three varieties, although Melody^TM^ displayed slightly higher values.

The new hybrid varieties presented higher phenolic contents (52.4–187.3 mg/100 g FW), surpassing Melody^TM^ in most cases due to the greater accumulation of anthocyanins and/or tannins. These values are consistent with those reported by [17] for colored grape varieties (17–250 mg GAE/100 g FW) and by [7] for table and wine grapes (103.1–257.0 mg/100 g FW), though differences may arise from the solvent and extraction methods used. In general, the hybrids showed phenolic concentrations comparable to or higher than those found in red or black wine grapes, confirming their potential as rich sources of bioactive compounds.

Among hybrids, RF04 had the highest anthocyanin concentration (337.7 mg/100 g FW), nearly ten times higher than Melody^TM^ (30.1 mg/100 g FW), though its tannin level (55.3 mg/100 g FW) was about half that of Melody^TM^ (109.5 mg/100 g FW). RF01, RF10, and RF11 exhibited the greatest tannin accumulation, approximately 2.5 times higher than Melody^TM^, and high anthocyanin levels (169.6–187.7 mg/100 g FW). RF02 stood out for its simultaneous high anthocyanin (261.3 mg/100 g FW) and tannin (216.5 mg/100 g FW) contents, resulting in the highest antioxidant activity (762.7 mg/100 g FW), even exceeding that of blueberries (230.8–435.5 mg/100 g FW). Conversely, RF05 showed the lowest anthocyanin (78.2 mg/100 g FW) and tannin contents (92.6 mg/100 g FW), corresponding to the lowest antioxidant activity (195.3 mg/100 g FW). Other hybrids exhibited intermediate levels of phenolics and antioxidant activity, suggesting a direct but not strictly proportional relationship between phenolic content and antioxidant response.

Tannin levels in all hybrids and Melody^TM^ were higher than those reported by [38] for Monastrell, Cabernet, and Syrah (28.2–72.1 mg/100 g FW) and close to those of some table grapes studied by [39] (39.6–95.6 mg/100 g FW). Anthocyanin contents were also generally higher than those reported by [7,40] for table grape varieties, supporting the hypothesis that darker hybrids accumulate larger amounts of these pigments, as previously related to grape coloration and antioxidant potential [17]. However, as noted by these authors, the association between anthocyanin content and antioxidant activity is not always linear, possibly due to the presence of other antioxidants such as ascorbic acid or variations in phenolic composition that modify the overall antioxidant response [41].

In this study, no clear correlation was observed between anthocyanin or tannin levels and total phenolics measured by the Folin method, nor with antioxidant activity. This may be attributed to the presence of other minor phenolics such as flavonols and stilbenes not quantifiable by spectrophotometry but contributing synergistically to antioxidant capacity [42]. Therefore, detailed phenolic profiling is essential to understand how specific compounds and their interactions determine the antioxidant potential of these grape varieties.

### 3.2. Concentration and Composition of Phenolic Compounds by HPLC

In general all hybrids have shown a significant increase in the content of phenolic compounds; however, the determination of the concentration of each of the compounds that constitute the different phenolic families was only carried out for those hybrids that presented the best agronomic behavior (good fertility index) and the best sensory characteristics (firm texture, large size and good flavor). Based on that, RF03, RF04, RF05, RF06 and RF12 were selected.

The anthocyanin concentrations obtained by HPLC were consistent with those measured by spectrophotometry (Table 2), ranging from 33.3 to 166.4 mg/100 g FW. RF04 presented the highest anthocyanin content, followed by RF03, while RF05 showed the lowest. These values exceed those previously reported for several table grape varieties such as Flame, Red Globe, Napoleon, and Crimson [16] and are even higher than those found in some red wine grapes, including Monastrell, Cabernet Sauvignon, Syrah, and Merlot [43,44]. This confirms the strong pigmentation potential of the selected hybrids.

Regarding anthocyanin profiles, RF04 was dominated by non-acylated monoglycosides, while RF06 showed a predominance of acylated forms; the remaining hybrids exhibited balanced proportions of both types of anthocyanins. Malvidin-3-glucoside was the major pigment in RF03, RF06, and RF12, similar to wine grapes [16,43,45], whereas peonidin-3-glucoside predominated in RF04 and RF05. These findings are consistent with [40], who reported higher malvidin-3-glucoside and acylated anthocyanin levels in black grapes, while red grapes showed greater proportions of peonidin-3-glucoside. Similar patterns have been observed in both seedless and colored table grapes [16,46] and in Garnacha Tintorera, where peonidin-based anthocyanins dominate the flesh and malvidin-based compounds the skin [41]. Flavonol concentrations were lower than those of anthocyanins, ranging between 2.09 and 7.84 mg/100 g FW. RF04 again exhibited the highest content, followed by RF03 and RF12, while RF05 showed the lowest. The major flavonols were myricetin-3-glucoside, quercetin-3-glucoside, and isorhamnetin-3-glucoside, although their relative abundance varied among hybrids. Compared with literature values [47], the hybrids in this study presented higher flavonol contents, particularly for quercetin derivatives, which are typical of both table and wine grapes [48,49,50,51,52,53,54,55,56]. These differences may result from genotypic variation or environmental conditions affecting flavonol biosynthesis.

Stilbene concentrations ranged from 781.3 to 3913.4 µg/100 g FW, with RF06 and RF03 showing the highest and lowest values, respectively. The predominant compounds were trans-resveratrol, cis-piceide, and viniferine, though cis-resveratrol was absent in some hybrids. In RF05, RF06, and RF12, trans-resveratrol predominated, consistent with previous findings showing variability in stilbene composition among grape types [17,49]. Such variability has been attributed to factors including the stage of ripening, the grape varieties or various external stimuli [51,52].

These results show that most of the hybrids analyzed have great phenolic potential; moreover, when these results are compared to those observed for blueberries, one of the berries that have raised considerable interest in consumers in view of their high content in bioactive compounds, we found values for total tannin content of 160 mg/100 g [53], while the total anthocyanin content ranged between 166.0 and 226.6 mg/100 g [54,55], being very close to those obtained in this study for the different hybrids. Moreover, the anthocyanin and flavonol profile shown by blueberries is completely different from that observed in grape hybrids, as they are mainly glycosylated with galactose and arabinose. Very high contents of malvidin-3-galactoside, and delphinidin-3-galactoside were reported by [56], while [55] reported predominant malvidin-3-glucoside, and at malvidin-3-galactoside and in lower concentration peonidine 3-galactoside, cyanidin 3-arabinoside and cyanidin 3-galactoside. The sum of flavonol glycosides obtained for different blueberry varieties by [57] varied between 13.7 mg/100 g for the Duke variety to 27.2 mg/100 g for the Simultan variety, quercetin being the most abundant in all varieties, while the sequence in terms of the presence of other flavonols was variety-dependent. The authors also reported that the most significant conjugate sugar in blueberry extracts was galactoside (ranging from 35.8% to 72.1%). These values of flavonols are higher than those presented by different hybrids. In the case of the stilbenes, only a content of 0.4 mg/100 g FW for trans-resveratrol was reported by [58].

The hybrids characterized by HPLC, together with the grape varieties Melody^TM^, Krissy^TM^ and Timpson^TM^, were selected for study in intestinal transport/absorption after digestion by comparing them to white grapes and pineapple as reference fruits, since their consumption is recommended in low-calorie diets and for controlling blood glucose levels because they have a moderate glycemic index (around 59–66), which prevents postprandial blood sugar levels, due to their moderate soluble fiber content that helps reduce the sugar absorption in the intestine; their insoluble fiber, which promotes healthy intestinal transit, by containing the enzyme bromelain, which has anti-inflammatory, anti-edema, and potential digestive benefits and antioxidant compounds such as vitamin C, some flavonols and phenolic acids [59].

### 3.3. Biocompatibility of the Digests in the Intestinal Model to Study Intestinal Transport

To better simulate the physiological conditions found in the body, first, the fruits were subjected to a dynamic gastrointestinal digestion simulation. This approach allows for a more realistic assessment of the changes in glucose content during the digestive process. A comparative analysis was performed to measure the glucose levels in the fruits (expressed as g of glucose per 100 g of fruit) both before digestion and in the soluble fraction obtained after digestion.

Then, the potential functional effect of metabolites and compounds that could pass into the bloodstream after digestion and intestinal absorption was studied. As a preliminary step, the application dilution or biocompatible concentration was established through a cytotoxicity study using the soluble fraction (SF) obtained after in vitro digestion. Results are shown in Figure 2.

The results show that the 1/20 dilution of the SF of the different grape varieties is the dilution that allows for a maximum biocompatible concentration of all the samples. In addition, controls including pineapple and blank digesta were evaluated and showed similar results. This finding highlights the potential of these grape varieties and hybrids to be processed and ingested without adverse effects on intestinal cells, which is crucial for evaluating their functional health benefits.

Once the conditioning and non-toxic concentration of the gastrointestinal digests to be applied was determined, the intestinal transport test was carried out, following the methodology indicated in Section 2.

### 3.4. Changes in Glucose Transport in the Intestinal Transport Model

After an incubation period of 1 h, the basal medium was collected for its corresponding glucose analysis. The glucose levels in the absorbed medium allowed the calculation of the % of glucose transport. In addition, a comparison has been made of the intestinal transport of glucose from grape samples with respect to a reference fruit in low-calorie diets, such as pineapple. Results regarding % of transported glucose are shown in Figure 3.

The results show that the percentage of intestinal glucose transport is lower when using the SF of all grape varieties and hybrids compared to the SF of pineapple. Furthermore, all the hybrids studied exhibited reduced levels of transported glucose in comparison to the three commercial grape varieties, highlighting the improvement in this parameter.

Since all the samples analyzed showed similar amounts of glucose, the differences in the % of glucose may be associated with the grape phenolic compounds, which are present in a higher content in hybrids compared to grape varieties. Therefore, this suggests that these hybrids may offer better control over postprandial glucose levels. Moreover, the results of theoretical glucose are shown in Figure 4.

If the theoretical g of glucose transported is calculated considering the % of transported glucose, all grape varieties and the hybrid RF06 presented glucose levels comparable to those of reference fruit (pineapple), while lower levels were shown by the rest of the hybrids. This suggests that these hybrids may have a lower glycemic impact than pineapple, which is beneficial for metabolic health and insulin sensitivity.

Since all the samples analyzed contained similar amounts of glucose, the observed differences in glucose transport are likely attributable to the phenolic compounds present in the grapes. Phenolic compounds, known for their bioactive properties, can interact with intestinal transporters such as SGLT1 (sodium–glucose-linked transporter 1) and GLUT2, potentially modulating glucose absorption. Some in vitro studies have demonstrated the ability of phenolic compounds to influence glucose transport at the cellular level. Specifically, anthocyanins and tannins have been shown to inhibit α-glucosidase and α-amylase, which are involved in the breakdown of carbohydrates into glucose, suggesting that they can slow down the absorption of glucose in the digestive system [60,61]. A positive correlation between tannin size and structural complexity to bind with enzymes and inhibit their activity (causing a reduction in the glycemic response of the individual) was previously reported [62], while a higher susceptibility of the α-glucosidase and α-amylase activities to glucoside anthocyanins and non-glucoside compounds, respectively, was suggested by [63], although the highest inhibitory activity was reported in acylated anthocyanins [64], which are not present in all berries, possibly due to their greater stability in the gut [65].Another inhibitor is resveratrol (stilbene) capable of decreasing the affinity of α-amylase for the substrate by competitive inhibition with the participation of its structural hydroxyl groups [66,67], while the activity of the α-glucosidase enzyme is non-competitively inhibited by the compound piceatanol [66]. Flavonols such as rutin, quercetin and kaempferol have also shown the ability to inhibit these enzymes responsible for carbohydrate digestion [22].

Anthocyanins, flavonols and stilbenes may also enhance the activity of glucose transporters, blocking sodium/glucose cotransporter 1 (sGLT1)- and glucose transporter type 2 (GLUT2)-mediated intestinal glucose absorption, improve peripheral glucose transporter type 4 (GLUT4) function, and increase mitochondrial density (Solverson, 2020). Moreover, these families of phenolic compounds have been shown to activate AMPK (AMP-activated protein kinase), a key regulator of energy metabolism, which improves final glucose uptake in insulin-sensitive tissues [68]. The structure of anthocyanins appears to significantly influence their ability to modulate glucose transport. Variations in hydroxylation, methoxylation, and glycosylation can affect both their bioavailability and their interaction with glucose transporters and signaling pathways [69]. Thus, [70] reported a greater cyanidin-3-glucoside and -galactoside, and peonidin-3-galactoside relative abundance in plasma and the majority of arabinoside containing anthocyanins were poorly absorbed when compared to their abundance in the bilberry extract, which contained 36% anthocyanins; and 15 different forms were identified, including cyanidin, peonidin, delphinidin, petunidin, and malvidin, each with arabinose, galactose, and glucose. On the other hand, [71] indicated that the uptake study using strawberry and red grape anthocyanin extracts constituted mainly by 90% of cyanidin-3-glucoside, pelargonidin-3-glucoside and pelargonidin-3-rhamnosylglucoside in the case of strawberry and 39.76% of delphinidin-3-glucoside, cyanidin-3-glucoside, petunidin-3-glucoside, peonidin-3-glucoside, and malvidin-3-glucoside in the case of grape demonstrated that red grape anthocyanins are less likely to be taken up by cells than those of strawberry (0.015% versus 0.073%) suggesting that uptake of anthocyanins in red grape are less favorable than those in strawberry, which indicates the effect of anthocyanin structure on the extent of the uptake into cells. These same authors reported that amongst the anthocyanins present in the strawberry extract, cyanidin-3 glucoside was the most taken up by cells at the same initial amount of anthocyanin, and the presence of rhamnosyl–glucoside to pelargonidin seems to favorably increase the uptake compared to glucoside, and the presence of glucose in the red grape extract decreased the uptake of anthocyanins by potential competition between glucose and anthocyanins for uptake into cells through the same transporter. This same uptake mechanism was also reported for flavonol glycosides and resveratrol [72,73,74]. Interestingly, not only the native anthocyanins present in the food may interact with glucose transporters, but also their metabolites, as was demonstrated by malvidin-3-O-glucuronide [75]. This contributes to the antidiabetic effects to these families of phenolic compounds.

This suggests that the hybrids can have greater potential for controlling postprandial glucose levels, which is a critical factor in managing glycemic responses and reducing the risk of type 2 diabetes. The findings underscore the importance of breeding programs focused on increasing the phenolic content in grape varieties as a strategy to enhance their functional properties and potential health benefits.

### 3.5. Changes in Insulin Resistance Biomarkers (Glycogen) in the Liver Model

The amount of glycogen generated is measured as an indicator of a good response to insulin, since glucose is converted into glycogen by the action of insulin. In organisms with type 2 diabetes, there is insulin resistance; the glucose present in the bloodstream is not transformed into glycogen (the pancreas does not produce enough insulin) and cannot be used as energy. The results are shown in Figure 5.

Results show that the intracellular level of glycogen, reduced when simulating insulin resistance in vitro (#), significantly changed after treatment with the absorbed fractions of RF03, RF05, and RF06, or tended to change in RF12 compared to the digestion blank. The rest of the varieties do not significantly modify this biomarker compared to the digestion blank, and no changes were observed in the induced physiological state.

Considering changes in the intermediate biomarker (akt), results do not show significant change in any case, despite being observed in the finalist biomarker, so it could indicate that it is not one of the intermediaries involved in the route that improves glycogen reserves.

Following the selection of varieties during the project, an improvement in insulin resistance has been observed compared to commercial varieties, analyzed using an in vitro model.

An increase in glycogen synthesis in HepG2 cells treated with anthocyanin-rich extracts from blueberries and blackberries has been shown by [76,77] to elevate the activity of glycogen synthase, an enzyme critical for glycogen formation, and to increase glycogen content in these cells. The tannins may also enhance glycogen synthesis in HepG2 cells by modulating key signaling pathways. So, tannin-rich extracts from grape seeds have been shown to increase the phosphorylation of glycogen synthase kinase 3β (GSK3β), leading to a promotion of glycogen synthase activity [78]. Resveratrol, a stilbene found in red wine and certain berries, has been reported to promote glycogen synthesis by activating the sirtuin 1 (SIRT1) pathway. SIRT1 activation improves insulin sensitivity, leading to an enhanced glycogen storage capacity in HepG2 cells [79,80]. Some studies indicate that flavonols, particularly quercetin and kaempferol, activate AMPK (AMP-activated protein kinase), a regulator of cellular energy metabolism, which promotes glucose uptake and glycogen synthesis [81]. Moreover, all phenolic compounds exhibited antioxidant properties, reducing radical oxygen species (ROS) levels and increasing the expression of antioxidant enzymes, which may contribute to its beneficial effects on insulin resistance [82].

The results also showed that among the hybrids studied in this work, RF03, RF05 and RF06, presented an effect on the modulation of one of the two biomarkers analyzed, related to insulin resistance [83]. Both biomarkers have been reported to be modulated in vitro by different compounds isolated from grapes [84], although none of them have evaluated this effect of the whole fruit, or after simulating its digestion or absorption in vitro. Hybrids have a higher content of phenolic compounds and antioxidant capacity than grape varieties, being responsible for a greater reduction in glucose absorption, as well as insulin response. The lower effect on glycogen intracellular level by hybrids RF4 and RF12, which show a higher content of phenolic compounds than commercial grape varieties and the different values found in the concentration and composition of anthocyanins, tannins, flavonols and stilbenes among hybrids RF03, RF05 and RF06 that have shown a positive effect on the glycogen biomarker, highlights synergistic effects between some of the phenolic compounds present in these hybrids and even between the phenolic compounds with the dietary fiber present in each of them, which can inhibit digestive enzymes involved in the breakdown of carbohydrates into glucose due to the formation of enzyme–polyphenol–fiber complexes [85,86] and their content could differ between them. Although the fiber content was not measured in this study, it is important to note that hybrid grapes developed through crosses involving wine grape varieties may exhibit higher dietary fiber content than traditional table grapes, as wine grapes are characterized by thicker skins rich in insoluble dietary fiber, whereas table grapes typically have thinner skins, resulting in lower fiber content [87,88].

## 4. Conclusions

This study demonstrates that the newly developed seedless table grape hybrids with red flesh and black skin possess significantly higher phenolic content and antioxidant capacity than commercial grape varieties. In vitro models of intestinal absorption and hepatic insulin resistance revealed a reduced glucose transport rate and enhanced glycogen accumulation, indicating potential improvements in insulin sensitivity. These results underscore the scientific and practical relevance of these hybrids as promising candidates for supporting postprandial glycemic control and for use in breeding programs targeting fruit with enhanced functional properties. Nevertheless, further research, including in vivo and clinical studies, is warranted to validate these effects and elucidate the bioavailability and metabolic fate of their bioactive compounds.

## Figures and Tables

**Figure 1 foods-14-04035-f001:**
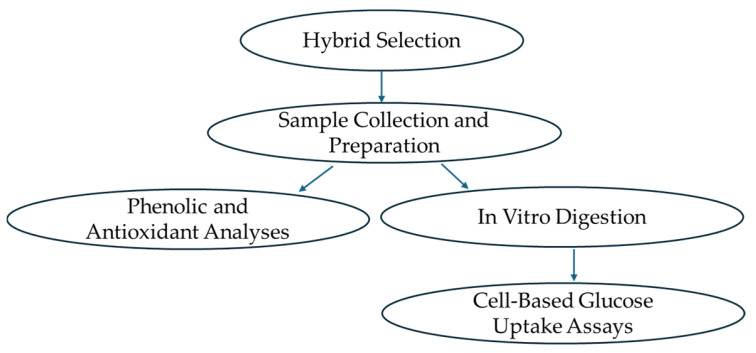
Roadmap of sample analysis.

**Figure 2 foods-14-04035-f002:**
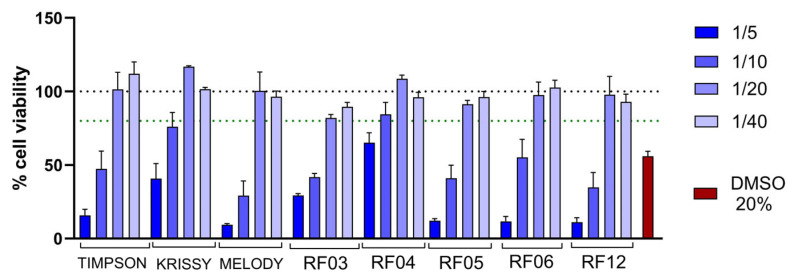
Biocompatibility of SF from gastrointestinal digests of the grape varieties under study in intestinal cells (Caco-2). Cell viability was assessed after 1 h exposure to the indicated dilutions of SF from the grape varieties. The black dotted line represents 100% cell viability, while the green line indicates the 80% viability threshold. Data are normalized to untreated cells, set to 100. Results are presented as mean ± SEM (*n* ≤ 5). The selected dilution was used in subsequent experiments. DMSO was included as a positive cytotoxic control.

**Figure 3 foods-14-04035-f003:**
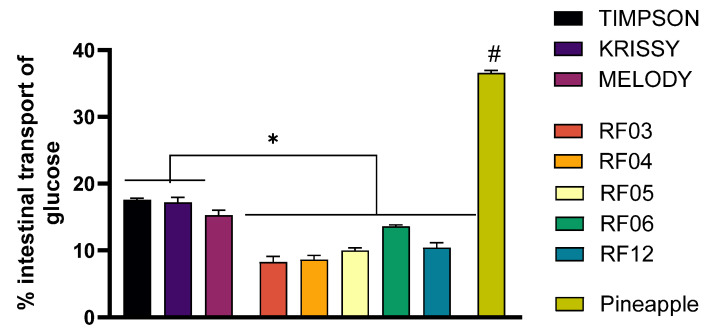
Evaluation of glucose transport percentage of SF from the grape varieties under study and the reference fruit. Glucose transport was assessed after 1 h exposure to diluted SF from the grape varieties in the transepithelial intestinal transport study. Results are presented as mean ± SEM (*n* ≤ 4). Pineapple was used as the commercial fruit reference. (# *p* < 0.05) indicates comparison with the reference fruit; (* *p* < 0.05) indicates comparison between grape varieties. Statistical analysis was performed using a two-tailed Student’s *t*-test.

**Figure 4 foods-14-04035-f004:**
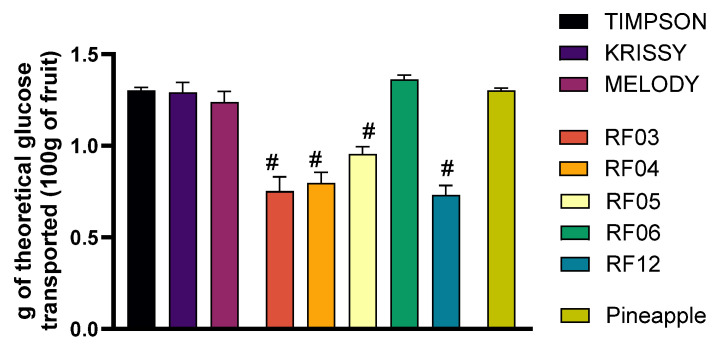
Evaluation of the theoretical amount of glucose absorbed after ingestion of the grape varieties under study and the commercial pineapple. Grams of theoretical glucose were calculated after 1 h exposure to diluted SF from the grape varieties in the transepithelial intestinal transport study. Results are presented as mean ± SEM (*n* ≤ 4). Pineapple was used as the commercial fruit reference. (# *p* < 0.05) indicates comparison with the reference fruit. Statistical analysis was performed using a two-tailed Student’s *t*-test.

**Figure 5 foods-14-04035-f005:**
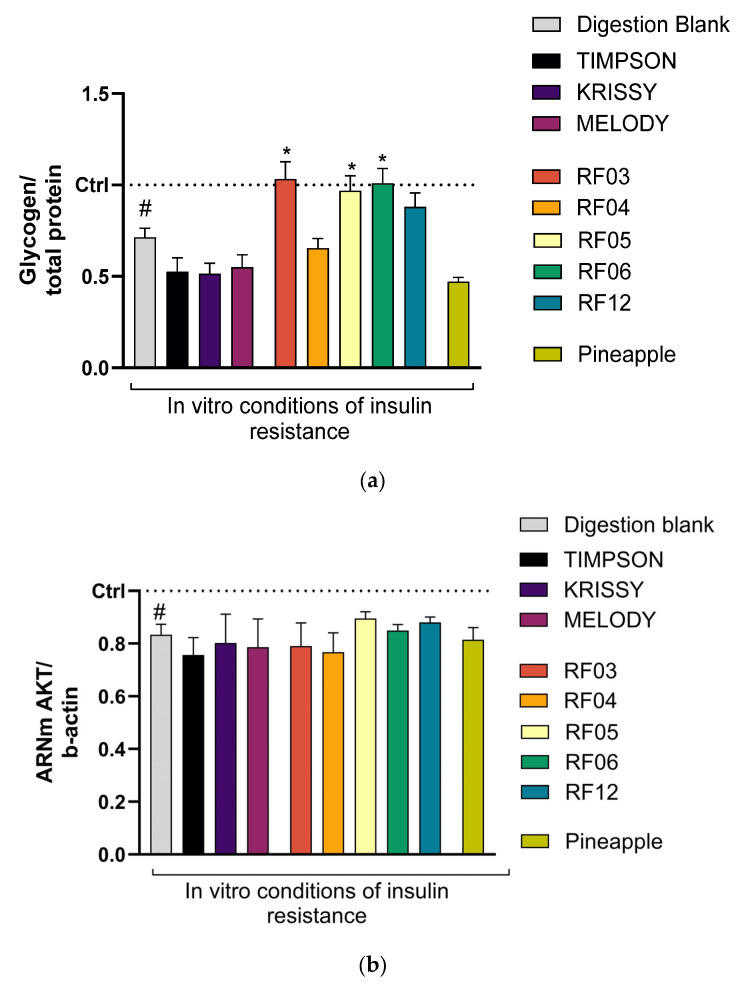
Changes in biomarkers associated with the liver model to study modulation of insulin resistance. (**a**) Changes in the AKT biomarker and (**b**) changes in glycogen, both associated with insulin resistance, were evaluated after inducing this condition in vitro in hepatic cells and exposing them to samples obtained from the intestinal absorption model. Data are normalized to untreated control cells (Ctrl), set to 1 and indicated by a black dotted line. Results are presented as mean ± SEM (*n* ≤ 4). Pineapple was used as the commercial fruit reference. (# *p* < 0.05) indicates comparison with the control without insulin resistance induction; (* *p* < 0.05) indicates comparison with the digestion blank. Statistical analysis was performed using a two-tailed Student’s *t*-test.

**Table 1 foods-14-04035-t001:** Spectrophotometric parameters measured in grape samples.

Samples	TEAC	TFC	TA	TT
Timpson^TM^	58.3 ± 2.9 a	21.1 ± 1.0 a	0.0 a	38.6 ± 1.9 a
Krissy^TM^	62.4 ± 3.1 a	29.8 ± 1.5 b	9.2 ± 0.5 b	33.6 ± 1.7 a
Melody^TM^	80.9 ± 4.0 a	61.0 ± 3.1 d	30.1 ± 1.5 c	109.5 ± 5.5 cd
RF01	660.6 ± 33.0 h	161.6± 8.1 i	169.6 ± 8.5 h	250.8 ±12.5 h
RF02	762.7 ± 38.1 i	164.1 ± 8.2 i	261.3 ±13.1 k	216.5 ± 10.8 g
RF03	259.3 ± 12.9 c	113.5 ± 5.7 h	180.6 ± 9.0 ij	181.6 ± 9.1 f
RF04	295.0 ± 14.7 d	187.3 ± 9.3 k	337.7 ± 16.9 l	55.3 ± 2.7 b
RF05	195.3 ± 9.7 b	52.4 ± 2.6 c	78.2 ± 3.9 d	92.6 ± 4.6 c
RF06	261.2 ± 13.1 c	73.8 ± 3.7 e	150.2 ± 7.5 g	111.3 ± 5.5 d
RF07	329.6 ± 16.5 e	84.0 ± 4.2 j	123.8 ± 6.2 f	154.1 ± 7.7 e
RF08	333.4 ± 16.7 e	75.0 ± 3.8 e	189.3 ± 9.4 j	119.9 ± 5.9 d
RF09	230.8 ± 11.5 c	74.5 ± 3.7 e	177.0 ± 8.8 hi	114.3 ± 5.7 d
RF10	435.5 ± 21.8 g	99.3 ± 4.9 g	187.7 ± 9.4 j	249.3 ± 12.4 i
RF11	372.0 ± 8.6 f	88.4 ± 4.4 f	101.1 ± 5.0 e	240.1 ± 12.0 i
RF12	257.6 ± 12.9 c	69.2 ± 3.5 de	123.8 ± 6.2 f	194.5 ± 9.7 g

Abbreviations: TEAC: antioxidant capacity, TFC: total phenolic compound, TA: total anthocyanin, TT: total tannin. The concentration of all parameters shown in the table was expressed as (mg/100 g FW). Different letters in the same column indicate significant differences between samples by LSD test (*p* ≤ 0.05, *n* = 3).

**Table 2 foods-14-04035-t002:** Phenolic compounds in grape samples by HPLC.

Compounds	RF03	RF04	RF05	RF06	RF12
*Anthocyanins (mg*/*100 g FW)*					
Delphinidin-3-Glu	1.9 ± 0.1 c	11.± 0.6 d	0.4 ± 0.0 ab	0.1 ± 0.0 a	0.9 ± 0.0 b
Cianidin-3-Glu	0.5 ± 0.0 a	8.0 ± 0.4 c	0.4 ± 0.0 a	0.9 ± 0.0 b	0.3 ± 0.0 a
Petunidin-3-Glu	3.0 ± 0.1 c	10.9 ± 0.5 d	0.6 ± 0.0 a	0.4 ± 0.0 a	1.7 ± 0.1 b
Peonidin-3-Glu	9.9 ± 0.5 c	47.4 ± 2.4 d	10.7 ± 0.5 c	2.0 ± 0.1 a	5.9 ± 0.3 b
Malvidin-3-Glu	20.8 ± 1.0 c	42.3 ± 2.1 d	6.2 ± 0.3 a	12.1 ± 0.6 b	13.7 ± 0.7 b
TA-Acylated	40.3 ± 2.0 cd	45.9 ± 2.3 d	15.1 ± 0.7 a	36.6 ± 1.8 c	23.3 ± 1.1 b
ƩTotal	76.4 ± 3.8 c	166.4 ± 8.3 d	33.3 ± 1.6 a	52.1 ± 2.6 b	45.7 ± 2.3 ab
*Flavonols* *(mg*/*100 g FW)*					
Myricetin-3-Glu	0.92 ± 0.04 c	2.71 ± 0.13 d	0.23 ± 0.01 a	0.65 ± 0.03 b	0.52 ± 0.02 b
Quercetin-3-Glur	0.41 ± 0.02 c	0.24 ± 0.01 b	0.17 ± 0.01 a	0.28 ± 0.01 b	0.92 ± 0.04 d
Quercetin-3-Glu	0.43 ± 0.02 a	2.15 ± 0.01 d	0.88 ± 0.04 c	0.55 ± 0.03 b	0.51 ± 0.02 b
Laricitrin-3-Glu	0.38 ± 0.02 c	0.46 ± 0.02 d	0.14 ± 0.01 a	0.29 ± 0.01 b	0.35 ± 0.02 c
Isorhamnetin-3-Glu	0.41 ± 0.02 b	1.39±0.07 d	0.53 ± 0.02 c	0.48 ± 0.02 c	0.32 ± 0.01 a
Siringetin-3-Glu	0.55 ± 0.03 d	0.52 ± 0.02 d	0.16 ± 0.01 a	0.28 ± 0.01 b	0.49 ± 0.02 c
ƩTotal	3.09 ± 0.15 c	7.58 ± 0.38 d	2.09 ± 0.10 a	2.54 ± 0.13 b	3.11 ± 0.15 c
*Stilbenes* *(µg*/*100 g FW)*					
Trans-Piceid	12.2 ± 0.6 ab	35.7 ± 1.8 c	10.1 ± 0.5 a	14.5 ± 0.7 b	8.8 ± 0.4 a
Piceatannol	59.1 ± 2.9 b	190.1 ± 9.5 c	1.5 ± 0.1 a	4.9 ± 0.2 a	4.1 ± 0.2 a
Cis-Piceid	269.4 ± 13.5 c	316.2 ± 15.8 d	55.5 ± 2.8 a	112.2 ± 5.6 b	88.4 ± 4.4 ab
Trans-Resveratrol	243.9 ± 12.2 a	384.5 ± 19.2 a	1081.6 ± 54.1 b	2480.2 ± 124.0 d	2017.8 ± 100.9 c
Cis-Resveratrol	0.0 a	0.0 a	535.2 ± 26.7 b	1220.8 ± 61.0 d	994.2 ± 49.7 c
Viniferin	197.2 ± 9.8 c	204.3 ± 10.2 c	62.0 ± 3.1 a	80.8 ± 4.0 a	144.7 ± 7.2 b
ƩTotal	781.8 ± 39.1 a	1130.8 ± 56.5 b	1745.8 ± 87.3 c	3913.4 ± 195.7 e	3258.0 ± 162.9 d

Abbreviations: Glu: glucose, Glur: glucuronide, TA: total anthocyanins. Different letters in the same row indicate significant differences between samples by LSD test (*p* ≤ 0.05, *n* = 3).

## Data Availability

The original contributions presented in the study are included in the article, further inquiries can be directed to the corresponding authors.

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
