# Peer review of "Development of New Red-Fleshed Seedless Table Grapes: In Vitro Insights on Glucose Absorption and Insulin Resistance Biomarkers"

_foods, 2025, doi:10.3390/foods14234035_

Round 1

Reviewer 1 Report

Comments and Suggestions for Authors

This manuscript, entitled "Development of New Red-Fleshed, Seedless Table Grapes with Enhanced Functional Properties," presents a study with potential applications in the field of functional foods, supported by a comprehensive experimental design. However, areas for improvement remain in data presentation, structural organization, and writing precision.   While commercial varieties (Timpson, Krissy, Melody) were included as controls, the absence of a positive control (e.g., an extract with known hypoglycemic effects) limits the interpretability of the results.   The authors suggest that dietary fiber may influence glucose absorption, yet no data on fiber content are provided. It is recommended that this be measured and reported.   Figures lack error bars in several instances, and p-values are omitted for some results, which hinders the assessment of statistical significance.   The use of a dynamic digestion system combined with Caco-2/HepG2 cell models, alongside spectroscopic and HPLC-MS analyses, lends strong credibility to the compositional data.   The selection of only hybrids with "favorable agronomic and sensory characteristics" for detailed HPLC analysis may introduce bias. Clarifying the specific selection criteria would strengthen the methodology.   The introduction could be more concise, with a clearer and earlier articulation of the study's objectives. The Results section incorporates extensive literature comparisons, which would be better placed in a dedicated Discussion section, where comparisons with the present findings should be more thoroughly elaborated.   Figures 1–4 lack sufficiently detailed titles and captions. In tables, the notation that "different letters indicate significant differences" should specify the statistical test used (e.g., LSD).   Inconsistent formatting was observed, particularly in units (e.g., alternating use of "mg/100 g" and "mg/kg") and decimal places (varying between one and two digits). These should be standardized throughout.   Citation formatting is inconsistent, with some references missing in the text and others lacking complete details (e.g., year or volume/issue).   Overall, the discussion is somewhat fragmented. The manuscript would benefit from a consolidated discussion section, standardized units and reference formatting, and optimized figures and tables. Streamlining the introduction, focusing the discussion, and adding a concluding synthesis would significantly enhance the manuscript's clarity and impact.

Comments on the Quality of English Language

Several language issues were noted: for instance, "apport" should be replaced with "provide," and "stood out for its lower content" is better phrased as "had the lowest content." Some sentences are overly long and structurally loose; breaking these into shorter, more focused sentences would improve readability.

Author Response

Please, see attached letter.

Reviewer 2 Report

Comments and Suggestions for Authors

The manuscript presents an interesting study on new red-fleshed, seedless grape varieties and their potential effects on glucose absorption and insulin resistance. However, requires substantial revisions to improve its scientific rigor, methodological clarity, and data presentation.

  1. Please clearly indicate all groups in the sample preparation to facilitate reproducibility and proper interpretation of the results.
  2. The author should include the exact formula used for TPC determination, specify the calibration curve and expression unit, and the same question in Sections 2.3 Total anthocyanins, 2.4 Total tannins, and 2.5 Antioxidant capacity. : Please include the exact calculation formulas, reference standards, and units used to ensure reproducibility and comparison with other studies.
  3. Please clarify how grape samples were stored before analysis. If frozen, discuss whether repeated freeze–thaw cycles could affect polyphenol stability.
  4. The author should clarify in the Description of HPLC–DAD and HPLC–MS Analyses

The analytical method descriptions for anthocyanins and flavonols are ambiguous and may lead readers to believe that HPLC–DAD and HPLC–MS were performed on the same system.

  1. The Methods section describes RNA isolation and quantitative real-time PCR analysis of AKT gene expression, but there are no corresponding results or figures presented in the Results section. Please provide the qRT-PCR results.
  2. The current title is informative but overly long; the author should consider simplifying it to improve readability and focus.
  3. The manuscript currently lacks a clearly defined and well-structured conclusion. Please add a concise conclusion section that summarizes the key findings, highlights their scientific and practical significance, and indicates the limitations and future research perspectives.

Author Response

Please, see attached letter

Reviewer 3 Report

Comments and Suggestions for Authors

The consumption of low fruit diets significantly contributes to chronic diseases in humans. High polyphenol colored grape enhances their antioxidant, hypoglycemic, anti-inflammatory, anti-allergic, antibacterial, and anticancer properties. This paper reveals that seedless colored grape exhibit phenolic content, antioxidant activity, inhibit intestinal glucose absorption, and demonstrate potential in preventing diabetes onset. It not only show great development potential as colored grapes for human health, but also has important theoretical significance and high practical value for solving the increasingly serious problem of diabetes mellitus. The current manuscript is very interesting, basically clear and scientific, good organizational structure, as well as comprehensive analysis process. However, the logical hierarchy of some paragraphs need to be improved. Major revision can be published in Foods. However, there are some major issues need to be improved:

  1. Title: There are too many punctuation marks. Can they be removed, like “In Vitro Insights on Glucose Absorption and Insulin Resistance Biomarkers of Seedless Red Grape Development”.
  2. Abstract: Abstract Background information is abundant (lines 17 to 24), but the research results of this paper are few, so it should be rewritten, and reference can be made https://www.frontiersin.org/journals/plant-science/articles/10.3389/fpls.2025.1650803/full
  3. Introduction: Referees with low relevance and old references should be reduced, and recent relevant references in the past three years should be supplemented; The first five paragraphs can be refined to highlight the innovation of this paper.
  4. Materials and Methods: Overall text process, if it can be refined it is better; if it can be drawn a technical roadmap is easier to understand.
  5. Results: This paper has rich research content, and the results and discussion are divided into two parts; Secondary headings are used for many paragraphs in each section. It is recommended to use tertiary headings to make the paper more readable. Table 1 and Table 2 Can the sample calculate the overall mean and coefficient of variation; Leave as much blank space between paragraphs as possible, especially if you write strictly to the Foods template. Can the two small pictures in Figure 1 be merged into one large picture? Paragraphs with only one sentence should be merged; paragraph length can be adjusted.
  6. Discussion: This part needs to be rewritten; The note should include a discussion of five parts. Referees with low relevance and old references should be reduced, and recent relevant references in the past three years should be supplemented;
  7. Conclusion: This part is more detailed than the summary, but the two should be distinct and complementary
  8. References: Some old literature with high similarity can be simplified, Write strictly in the format of Foods and supplement relevant literature in the last five years.

Author Response

Please, see attached letter

Reviewer 4 Report

Comments and Suggestions for Authors

Studies demonstrated that fruits with high phenolic content might have the advantages provided by the antioxidant effect of phenolic compounds, by the improvement in glucose adsorption and by decreasing the glycemic index of the fruit.  In this paper the authors studied some of these new varieties of table grapes, which are not yet on the market. They evaluated their phenolic content and their antioxidant activity. In vitro methods were applied to evaluate their potential capacity to reduce the glucose uptake during intestinal absorption, as well as their effects on risk factor associated with diabetes development, such as biomarkers related to insulin resistance.

Some suggestions:

1.Lines 66-68, you wrote “Epidemiological evidence suggests that high consumption of these plant-based compounds may provide protection against coronary heart disease, stroke, and lung cancer [19]”. Why only against lung cancer? Please clarify.

2.Line 93 – please specify about which digestive enzymes it’s about.

3. Please reformulate the information written at lines 94-96. 

4. Line 100 – please specify which “metabolic parameters associated to this parameter”. 

5. Add please more details about “the new table grape varieties were obtained” (line 118). 

  1. Add please information concerning the stability and the solubility of the extract.
  2. Line 139 – what do you mean by “different extracts”? Please clarify.

8. Materials and Methods - it is not understood which are the tested extracts and how many extracts are tested. 

9. Point 2.6.2 Add please details concerning Estilbenes determination. 

10. Line 249, you wrote that “Dimethyl sulfoxide (DMSO) at 20 % was used as a positive cytotoxicity control.” Isn't that too much DMSO 20%? 

11. Lines 300-301: Which are the used primer sequences? 

Author Response

Please, see attached letter

Round 2

Reviewer 2 Report

Comments and Suggestions for Authors

The authors have responded and justified all comments. The paper can be accepted.

Reviewer 3 Report

Comments and Suggestions for Authors the manuscript has been sufficiently improved to agree publication in Foods
